# Changes in the Care Activity in Addiction Centers with Dual Pathology Patients during the COVID-19 Pandemic

**DOI:** 10.3390/jcm11154341

**Published:** 2022-07-26

**Authors:** Cinta Mancheño-Velasco, Daniel Dacosta-Sánchez, Andrea Blanc-Molina, Marta Narvaez-Camargo, Óscar Martín Lozano-Rojas

**Affiliations:** 1Department of Clinical and Experimental Psychology, University of Huelva, 21004 Huelva, Spain; cinta.mancheno@dpces.uhu.es (C.M.-V.); daniel.daco@dpces.uhu.es (D.D.-S.); andrea.blanc@dpces.uhu.es (A.B.-M.); marta.narvaez885@alu.uhu.es (M.N.-C.); 2Research Center on Natural Resources, Health and the Environment, University of Huelva, 21004 Huelva, Spain

**Keywords:** dual pathology, COVID-19, care activity, pandemic, drug addiction, mental health

## Abstract

Background: Health care provision during the COVID-19 pandemic and confinement has led to significant changes in the activity of addiction centers. These changes in healthcare activity may have had a greater impact on patients with dual pathology. The aim of this study is to compare the treatment indicators of patients with dual pathology in addiction centers during the pre-confinement, confinement, and post-confinement periods. Methods: A retrospective observational study was conducted for the period between 1 February 2019 and 30 June 2021. A total of 2785 patients treated in specialized addiction services were divided into three periods according to their time of admission: pre-confinement, confinement, and post-confinement. Results: During the pre-pandemic period, the addiction centers attended to an average of 121.3 (SD = 23.58) patients, decreasing to 53 patients during confinement (SD = 19.47), and 80.69 during the post-confinement period (SD = 15.33). The number of appointments scheduled monthly for each patient decreased during the confinement period, although this number increased after confinement. There was a reduction in the number of toxicological tests carried out both during and after confinement (except for alcohol). Conclusions: The results show a reduction in the number of patients seen and the care activity delivered to dual diagnosis patients. These results, which were caused by the COVID-preventive measures, may affect the progress and recovery of dual patients. A greater investment is needed to bring the care activity up to the standards of the years prior to confinement.

## 1. Introduction

The impact of the COVID-19 pandemic on mental health has been extensively documented [1,2,3,4]. Increased diagnoses of anxiety and depression have been described [5,6,7,8], as well as a rise in the number of suicides in the population [9]. In addition, several authors have pointed out that the consequences of the pandemic have been more negative for people who had previously been suffering from other mental disorders, including addictive disorders [10,11,12].

Some authors have reported that at the onset of confinement there was an increase in drug use, as reflected in indicators suggesting increased sales of alcohol [13,14] and cannabis [15]. Furthermore, higher rates of alcohol [16], cannabis, and other forms of drug abuse [17,18,19,20,21] have also been documented. For example, Chappuy et al. [19] reported a 29.2% increase in alcohol use, a 27.6% increase in cannabis use, a 36.2% increase in psychostimulant use, and a 25.9% increase in hypnosedative and opiate use. In addition, a 48.7% increase in behaviors associated with pathological gambling has also been detected. Other authors have reported increases in self-medication patterns in patients with opioid dependence [11] and a rise in overdose rates [22,23]. However, despite the above data, it should be borne in mind that consumption patterns may vary according to the country and the specific regulations in force during the pandemic [18,24].

In terms of health care provision, the pandemic, confinement, and the policy measures adopted by governments have led to significant changes in the care activities of specialized addiction centers. For example, Mark et al. [25] found a 28% decrease in admissions to treatment during the beginning of the pandemic compared to the previous year. In contrast, Aguilar et al. [26] noted an increase in care activity and higher relapse rates during the second half of confinement. In addition, other authors have reported changes in care patterns, with online appointments being prioritized and an increase in attendance at these appointments [27]. Likewise, it has been shown that confinement has led to an increase in the therapeutic needs of patients with addiction, with these patients also encountering more barriers to receiving therapeutic sessions and pharmacological treatments [11,28,29]. Some authors have also reported a slight increase in requests for pharmacological prescriptions by new patients, although an overall decrease in patients has also been noted [30]. On the other hand, Huskamp et al. [31] reported a decrease in the number of toxicological tests carried out in outpatient addiction centers.

These changes in healthcare activity may have had a greater impact on patients with dual pathology. Generally, these patients require more extensive follow-up due to the greater therapeutic complexity involved in comparison with patients without dual pathology [32,33]. In addition, the closure of some addiction centers [34] and the shift to virtual treatment have posed a major challenge to meeting the therapeutic needs of these patients. Therefore, some authors have warned of the worsening of comorbid mental disorders and disruptive behaviors both in confinement periods [35,36,37] and in the subsequent periods [38], in addition to a likely increase in relapses [39].

Although previous studies have suggested the potential impact of the pandemic on patients diagnosed with dual pathology, no studies have yet compared the treatment indicators of care activities implemented for patients with dual pathology in addiction centers across the pre-confinement, confinement, and post-confinement periods. Thus, the present study had the following objectives: (i) to examine the evolution of admissions to treatment for patients with dual pathology receiving coordinated care with mental health centers between February 2019 and June 2021; (ii) to analyze the sociodemographic profiles, consumption patterns, and psychopathological profiles of these patients; and (iii) to compare care indicators related to therapeutic appointments, toxicological tests, and treatment abandonment across the three specified time periods. As hypotheses based on those objectives, it is expected that:(a)The evolution of admissions to treatment decreased during confinement;(b)Patients with dual pathology who attend addiction care centers presented changes in their sociodemographic, consumption, and diagnosis profiles during the pandemic compared to the previous period;(c)Care indicators related to therapeutic appointments, toxicological tests, and treatment abandonment changed during the pandemic compared to the previous period.

## 2. Material and Methods

### 2.1. Design

Retrospective observational study for the period between 1 February 2019 and 30 June 2021. Patients were divided into three periods according to their time of admission to the addiction centers: pre-confinement (1 February 2019 and 15 March 2020), confinement (16 March 2020–31 May 2020), and post-confinement (1 June 2020–30 June 2021).

### 2.2. Participants

For the current study, we included only patients admitted between 1 February 2019 to 30 June 2021 in specialized addiction services with dual pathology. Inclusion criteria were the following: (1) to be older than 18 years of age, (2) to have at least one diagnosis according to the International Classification of Diseases 10 (ICD-10) of an addictive disorder (cocaine, heroin, alcohol, cannabis, or pathological gambling) and another comorbid mental disorder, and (3) to have a clinical indication to receive coordinated care with mental health services.

The final sample consisted of 2785 patients diagnosed with an addictive disorder and another mental disorder according to ICD-10. In addition, all patients of the sample had therapeutic prescriptions to receive care in mental health services according to the Ries [40] classification. This is a dimensional model based on the severity of the addictive disorder and other mental disorders. Depending on the severity levels of these disorders, patients receive treatment exclusively in mental health (severe mental disorder and mild addictive disorder), in addiction centers (severe addictive disorder and mild mental disorder), or in both services in a coordinated manner (severe mental health and addictive disorder). All patients in this study received coordinated care between specialized addiction centers and mental health units in Andalusia [41]. These patients were admitted to treatment in one of the 121 outpatient centers of the Public Network for Addiction Care in Andalusia (Spain). Of the sample, 1576 (56.6%) were admitted during the year prior to confinement, 160 (5.7%) were admitted during confinement, and 1049 (37.7%) were admitted to treatment from the end of confinement until 06/30/2021.

Most patients were male (74.8%), with a mean age of 40.4 years (SD = 11.69) at the time of admission to treatment. Most patients had completed primary (37.6%) or secondary education (23.5%). Regarding employment status, 22.7% of the patients were employed, 44.9% were unemployed, 25% were retired, 3.7% were studying, and 3.7% were in an unknown employment situation.

According to ICD-10 criteria, 37.6% of the patients were diagnosed with alcohol dependence or harmful use, 33.6% with cocaine, 22.3% with cannabis, 16.3% with opiates, and 3.2 with hypnosedatives. In addition, 4.5% of the patients were admitted for pathological gambling. Excluding tobacco addiction, 13.9% of these patients were diagnosed with dependent or harmful use of more than one drug.

### 2.3. Procedure

The data used in the present study belong to the EHR of the Information System of the Andalusian Plan on Drugs (SiPASDA). The EHR begins by recording information corresponding to the variables of the Treatment Demand Indicator (TDI) Standard Protocol 3.0 of the European Monitoring Centre for Drugs and Drug Addiction [42]. The TDI provides basic information on sociodemographic variables, drug use history, previous treatments, and infectious diseases at the start of treatment. In addition, clinical data collected during the periodic appointments that patients attend (with physicians, psychologists, nurses, and social workers) are incorporated into the clinical history of each patient. In these appointments, each team member (physicians, psychologists, nurses, and social workers) inputs the relevant patient information into the EHR. This information includes the diagnosis of the patients according to ICD-10 criteria, prescribed pharmacological treatments, psychological evaluation and treatments, results of toxicological tests, social status of the patient, and evolution of treatment. All this information is stored in a centralized database, and therapists can access the information at any time. Previous research conducted with this same data set has shown good reliability values [43].

Due to the pandemic, most of the Andalusian centers used telephones as the main channel for treatment admissions and follow-up. Critically ill patients received face-to-face care from professionals, while telephone services have been maintained for patient follow-up after the confinement period. The requests are recorded by health professionals in the Electronic Health Record (EHR).

### 2.4. Ethics and Approvals

This research has been approved by the Research Ethics Committee of the Andalusian Ministry of Health, who certified compliance with the requirements for the ethical handling of the information.

To access the EHRs, the researchers made a request to the General Secretary of Social Services of the Department of Equality and Social Policies of the Regional Government of Andalusia (Spain). Patients gave their authorization so the information could be registered in the system. The database is fully anonymized for both patients and professionals, so it is not possible to inform the participants about the study. The storage and encoding of this data comply with the General Health Law of 25 April 1986 (Spain) and Law 41/2002 of 14 November on patient autonomy, rights, and obligations regarding clinical information and documentation. It also complies with the Organic Law 3/2018 of 5 December 2018, regarding the protection of personal data and the assurance of digital rights, adapted to European regulations.

### 2.5. Statistical Analysis

The three groups were compared using nonparametric analyses, given the differences in sample size between the confinement group and the pre- and post-confinement groups.

The differences between qualitative variables were analyzed using Pearson’s chi-square test, and Cramer’s V statistic was used to calculate effect sizes. Quantitative variables were analyzed with the Kruskal–Wallis test, using the epsilon-squared test to calculate effect sizes.

Analyses were conducted using IBM SPSS Statistics software version 27 (Chicago, IL, USA).

## 3. Results

### 3.1. Evolution of Treatment Admissions between 1 February 2019 and 30 June 2021

Figure 1 shows the monthly evolution of the number of treatment admissions for each month analyzed, with respect to the patients receiving coordinated care with mental health services. This shows the downward trend in admissions of these patients. Thus, during the pre-pandemic period, the addiction centers attended to an average of 121.3 (SD = 23.58) patients with dual pathology per month, decreasing to 53 patients during confinement (SD = 19.47), and 80.69 (SD = 15.33) patients during the post-confinement period.

In percentage terms, the number of patients with dual pathology seen during the year prior to confinement was 7.2%, with this number increasing slightly during confinement (8.1%) and then falling to 6.7% in the year after confinement, and these differences were statistically significant (χ^2^ = 6.646; *d.f.* = 4; *p* = 0.036; V = 0.013). As shown in Table 1, the variations observed in these periods run parallel to the readmissions to treatment (patients requesting treatment who had previously been in treatment), with the highest percentage of readmissions to treatment occurring during confinement.

### 3.2. Sociodemographic Characteristics, Consumption Patterns, and Comorbid Diagnoses

Table 1 compares the three time periods according to sociodemographic variables, consumption patterns, and psychopathological diagnoses. There were no statistically significant differences in the sociodemographic profiles of the patients, although there was an increase in the number of women who were admitted to treatment during confinement. With respect to consumption patterns, it should be noted that treatment admissions for opiate use increased during confinement (and although the number of admissions subsequently decreased, the differences were statistically significant). Concerning admissions for patients with alcohol abuse/dependence, a slight decrease was observed during confinement, after which an increase of almost 5% was observed after confinement. However, admissions for cannabis dependence/abuse decreased after confinement. Finally, admissions for pathological gambling decreased during confinement, subsequently returning to pre-confinement levels.

Concerning the diagnoses of comorbid mental disorders, in general terms, no statistically significant differences were observed between the three periods, except for personality disorders. However, an increase in diagnoses of anxiety spectrum disorders was observed during confinement, mainly due to mixed anxiety-depressive disorders. On the other hand, a reduction in personality disorders diagnosed after confinement was observed. However, it should be borne in mind that after confinement, there was an increase in the number of patients with clinical indications for coordinated care with mental health services, although the diagnosis provided in the clinical history was generic (severe mental disorder-SMD-together with an addictive disorder of difficult clinical management).

### 3.3. Care Provision Indicators

Table 2 shows the care indicators for the three periods analyzed. With respect to the therapeutic sessions planned by the clinicians, the number of monthly appointments scheduled for each patient decreased during the confinement period, although this number increased after confinement. Regarding the care activity of the patients, it was observed that they attended a greater percentage of scheduled appointments during the confinement period, with no difference between pre-and post-confinement.

There was a reduction in the number of toxicological tests carried out both during and after confinement (except for alcohol). In the case of patients with alcohol-related problems, a greater number of tests were carried out after confinement. For the remaining substances, there was a significant reduction in the percentage of patients who underwent toxicological tests. It should be noted that of the five substances analyzed, a statistically significant increase in positive test results was only observed for opiates.

Concerning treatment retention, a significant reduction in the percentage of patients abandoning treatment was observed across the three periods.

## 4. Discussion

Various studies have shown how the pandemic has resulted in changes in the treatment demands placed on addiction centers and the healthcare provision patterns of clinicians [25,26,27,44], along with the associated impact on patients [35,36,37,38]. Unlike previous studies, this study focused exclusively on patients with dual pathology and analyzed the evolution of treatment admissions, profiles, and care indicators corresponding to the periods before, during, and after confinement, when various anti-COVID-19 measures were implemented in addiction and mental health services.

Concerning the first hypothesis, the present study has clearly shown a change in the evolution of treatment admissions of patients with dual pathology. Specifically, we have observed an increase in admissions during confinement followed by a drop in such admissions post-confinement. The increase in the number of patients admitted during confinement might be explained by treatment readmissions (patients who had previously been in treatment). This finding is similar to that of Di Lorenzo et al. [45]. Although these authors did not exclusively analyze patients with substance use disorders, they observed a reduction in urgent psychiatric consultations during confinement while this number increased in people who were already being treated. Therefore, the observed increase could be due to the fact that patients with pre-existing mental disorders experienced a marked deterioration of symptoms during this period. Concerning the decline in admissions post-confinement, other authors have reported a similar observation, and this may pattern be due to infection-control measures associated with COVID-19 [25,46].

With regard to our second hypothesis, we expected to find differences in the profiles of patients admitted across the three-time periods analyzed, a prediction that was not supported by our results. However, there was a notable percentage increase in women admitted to treatment during confinement. This may be due to the characteristic symptomatology of anxious-depressive disorder experienced during this stage since the percentage of women with this diagnosis increased from 24.9% before confinement to 41.4% during confinement. Other authors have also found that these emotional stress symptoms are more frequent in women [35,38]. Therefore, the symptomatology associated with this disorder is likely to be the factor that explains the percentage increase observed in this gender.

We also observed a significant increase in the number of patients admitted for opiate dependence. The reduced availability of opiates in the illegal market has possibly prompted patients dependent on this substance to come to addiction centers demanding pharmacological treatment [30]. However, barriers to obtaining epidemiological data on illicit drug use during the pandemic in Spain, especially for drugs such as opiates [47], make it difficult to test this hypothesis.

Concerning diagnoses of mental disorders, the results of the present study agree with those reported by other authors, indicating an increase in symptoms characteristic of mixed anxiety-depressive disorders during confinement [48]. However, we found no increase in the number of admissions to treatment in patients with personality disorders, which might be expected based on other studies [49]. In fact, quite the opposite trend was found—the number of admissions to treatment for these patients decreased after confinement. However, this decrease may be due to methodological problems associated with the data recording techniques since, as described above, there was a significant increase in patients without a specific ICD-10 diagnosis after confinement.

The analysis of our third hypothesis revealed that patients with dual pathology received less care during confinement, although some post-confinement indicators were similar to those observed pre-pandemic. Other authors have also reported this lower attendance to psychiatric services [50]. These observations may be due to the implementation of care protocols designed to protect these patients against COVID-19. However, despite this reduction in scheduled appointments, it was found that patients in treatment attended more appointments and showed a reduction in treatment dropout, in congruence with other studies conducted in addiction centers [44]. Thus, patients showed greater treatment adherence during confinement, although subsequently, care indicators showed activity equivalent to that of pre-confinement levels, with a notable reduction in treatment dropout. In addition, fewer toxicological tests were carried out during confinement, as reported by other authors [31], with no recovery of pre-confinement levels. It is likely that the risk of contagion associated with the collection of biological samples has influenced this reduction in care activity, with priority given to self-report measures of drug use.

We should consider some limitations to correctly interpret these findings and compare the results. One of the main aspects to consider is that patients receive treatment coordinated with mental health services. In this study, while the activity of addiction services has been analyzed, the activity of these patients in mental health services has not. Thus, we are observing only a part of the care provided to these patients without knowing the care indicators of these patients in mental health services. Previous studies conducted in patients with dual pathology under this care modality have shown that sometimes patients leave one of the care networks and remain in the other, depending on the addiction profile and psychopathological disorder of the patients [51,52]. Moreover, the present study was based on data obtained from the EHR registry. Although clinicians have been using EHRs in a standardized manner since 2015, the pressure of care experienced in the months studied herein could have produced slight errors in the completion of EHRs. This could explain, for example, the increase in patients without a specific ICD-10 diagnosis observed in the data. On the other hand, it is necessary to keep in mind that the study included patients with high severities of their respective addictive disorders and other mental disorders, and not only patients with other comorbid disorders. Consequently, it is likely that the prevalence of dual pathology observed in this study is lower than that observed in other studies of dual pathology conducted in addiction centers.

Despite these limitations, the present study provides useful information for understanding the changes produced by the COVID-19 pandemic. In particular, our results provide relevant knowledge about a large sample of patients with dual diagnosis and the health care provided in several addiction centers. As this is a coordinated treatment modality, we have observed only the care that has occurred in addiction centers and not the care that these patients have received in mental health centers. Bearing this in mind, the data have shown a reduction in the healthcare received by these patients. Moreover, it is striking that after confinement, the number of patients with dual pathology has decreased. Therefore, it is likely that there is a group of patients with dual pathology who are presently either only receiving care in mental health centers or are not attending health services. Thus, we suggest that the coordinated treatment modality followed by these patients with dual pathology has proven to be insufficient for providing adequate clinical care during the pandemic period. Therefore, we believe that it is now more necessary than ever to integrate mental health and addiction services for the coordinated treatment of these patients with dual pathology.

Future studies should continue to provide information on care activity and confirm the results found with these patients, so that these data can be used to inform the development of effective and efficient treatments for patients with dual pathology. In addition, future analyses could identify factors that may mediate and prevent some of the major risks in similar situations.

## 5. Conclusions

We can conclude that: (1) the period of confinement resulting from the coronavirus pandemic has triggered a reduction in the number of patients seen and the care activity delivered to dual diagnosis patients, including treatment admissions. At the end of the isolation period, the care activity of the addiction centers increased again. (2) There has been an increase in the number of patients admitted for opiate dependence and in reported symptoms characteristic of mixed anxiety-depressive disorders during confinement. (3) These results—due to the COVID-19 preventive measures—may impact the progress and recovery of dual patients. (4) A greater investment is needed to raise the current level of care up to the standards of the pre-pandemic period. (5) A precise evaluation of the impact of the pandemic on patients with dual pathology and care activity will require more time to analyze the full extent of its effects.

## Figures and Tables

**Figure 1 jcm-11-04341-f001:**
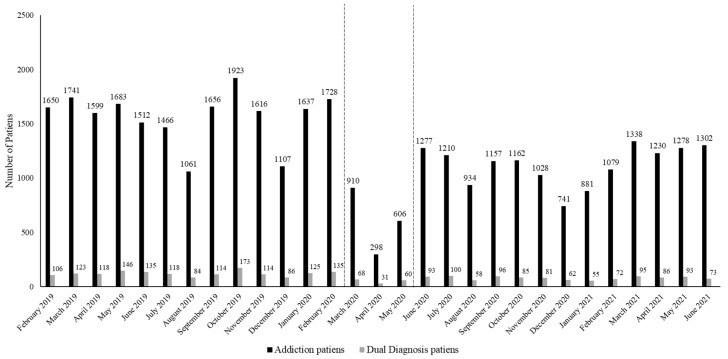
Evolution of patient admissions for treatment in the addiction centers.

**Table 1 jcm-11-04341-t001:** Sociodemographic characteristics, consumption profile, and diagnosis of patients with dual pathology.

	Admission Period			Statistics (*d.f*.)	*p*	Effect Size
	19 February–20 February	March–20 May	20 June–21 June			
No. of patients	1577 (56.6%)	159 (5.7%)	1049 (37.7%)			
% Patients (out of total patients)	7.2	8.1	6.7	χ^2^ (4) = 6.646	0.036 *	V = 0.013
Readmissions	67.4%	74.8%	63.2%	χ^2^ (4) = 10.549	0.005 **	V = 0.062
*Sociodemographic variables*						
Admission age (Mean, SD)	40.36 (11.536)	39.25 (11.698)	40.58 (11.920)	H (2) = 1.482	0.477	ε^2^ = 0.001
*Gender (%)*						
Male	75.8	72.5	73.8	χ^2^ (2) = 1.796	0.407	V = 0.025
Female	24.2	27.5	26.2			
*Educational level (%)*						
No education	17.0	14.1	14.6	χ^2^ (8) = 13.402	0.099	V = 0.049
Primary	39.1	37.8	35.3			
Secondary	22.3	22.4	25.6			
Baccalaureate/Degree	14.9	20.5	17.8			
Higher	6.7	5.1	6.8			
*Employment status (%)*						
Employed	22.7	18.5	23.3	χ^2^ (8) = 6.830	0.555	V = 0.035
Unemployed	44.9	45.9	44.8			
Retired	24.7	26.1	25.1			
Student	3.5	6.4	3.6			
Others	4.2	3.2	3.1			
*Main referral source (%)*						
Legal Services	3.1	4.4	2.5	χ^2^ (10) = 7.263	0.700	V = 0.036
Own initiative	41.9	48.1	42.2			
Family members	13.2	8.9	12.8			
Health Services	14.4	13.9	15.7			
Social Services	23.3	22.2	22.7			
Others	4.1	2.5				
*Variables related to consumption*						
Age of onset of consumption (Mean, SD)	19.64 (10.91)	20.81 (14.28)	19.74 (11.29)	H (2) = 0.739	0.691	ε^2^ = 0.000
*Admission drug (%)*						
Alcohol	36.4	34.4	39.3	2.930	0.231	V = 0.032
Cocaine	34.1	30.6	33.4	0.819	0.664	V = 0.017
Cannabis	23.4		20.2	4.447	0.108	V = 0.040
Opioids	18.3	21.3	12.5	18.840	0.000 **	V = 0.082
Hypnosedatives	2.9	4.4	3.5	1.455	0.483	V = 0.023
Pathological gambling	4.7	1.9	4.7	2.759	0.252	V = 0.031
*Other drugs used prior to admission (%)*						
Alcohol	55.6	55.0	58.0	χ^2^ (2) = 1.522	0.467	V = 0.023
Cocaine	36.8	35.0	39.4	χ^2^ (2) = 2.284	0.319	V = 0.029
Cannabis	39.9	40.6	37.4	χ^2^ (2) = 1.901	0.387	V = 0.026
Opioids	18.6	22.5	13.7	χ^2^ (2) = 14.231	0.001 **	V = 0.071
Hypnosedatives	8.1	6.3	7.9	χ^2^ (2) = 0.653	0.721	V = 0.015
*Frequency of consumption in the 30 days prior to admission (%)*						
Every day	44.1	45.2	42.4	χ^2^ (10) = 12.011	0.284	V = 0.048
4–6 days/week	7.4	5.1	9.1			
2–3 days/week	13.9	12.7	14.1			
1 day/week	5.5	10.8	5.9			
Less 1 day/week	7.8	6.4	8.2			
Did not consume	21.3	19.7	20.4			
*Variables related to the diagnosis of comorbid mental disorders*						
F 20. Schizophrenia, schizotypal disorders, and delusional disorders	16.2	16.4	16.2	0.004	0.998	V = 0.001
F 30–39. Mood disorders	17.4	13.8	16.9	1.297	0.523	V = 0.022
F 40–49. Neurotic, secondary to stressful situations, and somatoform disorders	31.9	34.0	32.6	0.362	0.834	V = 0.011
F 41. Mixed Anxiety-Depressive Disorder	16.8	20.1	17.6	1.241	0.538	V = 0.021
F 90. Hyperkinetic disorders	4.6	3.8	3.3	2.480	0.289	V = 0.030
Mental retardation	1.3	1.9	1.5	0.409	0.815	V = 0.012
*Adult personality and behavioral disorders (%)*						
Any personality disorder (F 60–F 60.9)	24.4	24.5	20.2	6.421	0.040 *	V = 0.048
F 60.0 and 60.1. Paranoid or schizoid personality disorder	2.0	2.5	1.3	2.220	0.330	V = 0.028
F 60.2–60.4. Antisocial, borderline, histrionic or narcissistic disorder	12.6	8.8	10.9	3.268	0.195	V = 0.034
F 60.5–60.7. Avoidance, dependence, or obsessive-compulsive disorder.	1.2	1.3	1.0	0.394	0.821	V = 0.012
F 60.9. Unspecified Personality disorder	8.8	11.9	7.2	4.688	0.096	V = 0.041
Patients without specified ICD-10 diagnosis	13.1	10.7	16.4	7.441	0.024 *	V = 0.052

Abbreviations: *d.f.*—degrees of freedom; SD—Standard Deviation; H—Kruskal–Wallis; V— Cramer’s V; * *p*-value ≤ 0.05; ** *p*-value < 0.01

**Table 2 jcm-11-04341-t002:** Care indicators for patients with dual pathology.

		Admission Period			Statistics (*d.f*.)	*p*	Effect Size
		19 February–20 February	March–20 May	20 June–21 June			
*Appointments (mean*, *SD)*							
Scheduled monthly		1.12 (2.22)	0.64 (0.74)	1.28 (2.32)	H (2) = 62.655	0.000 **	ε^2^ = 0.023
Percentage attendance		0.76 (0.23)	0.88 (0.25)	0.77 (0.25)	H (2) = 92.348	0.000 **	ε^2^ = 0.033
*Toxicological controls (mean*, *SD)*							
Alcohol	% Patients tested	7.7	0	10.4	χ^2^ (2) = 7.701	0.021 **	V = 0.086
	Average per patient	4.91 (4.88)	0	5.72 (14.82)	H (2) = 0.631	0.427	ε^2^ = 0.000
	Positive ratio	0.14 (8.75)	0	0.19 (0.35)	H (2) = 3.565	0.168	ε^2^ = 0.001
Cocaine	% Patients tested	53.6	24.5	45.4	χ^2^ (2) = 18.174	0.000 **	V = 0.139
	Average per patient	6.51 (12.75)	0.8 (1.14)	5.20 (5.83)	H (2) = 17.721	0.000 **	ε^2^ = 0.006
	Positive ratio	0.38 (0.40)	0.50 (0.58)	0.37 (0.41)	H (2) = 0.287	0.866	ε^2^ = 0.000
Cannabis	% Patients tested	52.3	20.0	40.1	χ^2^ (2) = 19.761	0.000 **	V = 0.178
	Average per patient	5.16 (6.16)	1.87 (0.64)	4.86 (4.59)	H (2) = 6.142	0.046 **	ε^2^ = 0.002
	Positive ratio	0.64 (0.42)	0.63 (0.52)	0.53 (0.45)	H (2) = 3.437	0.179	ε^2^ = 0.001
Opioids	% Patients tested	45.7	14.7	37.4	χ^2^ (2) = 13.031	0.001 **	V = 0.169
	Average per patient	4.75 (6.69)	1.60 (0.89)	5.04 (6.55)	H (2) = 2.825	0.243	ε^2^ = 0.001
	Positive ratio	0.30 (0.41)	0.75 (0.50)	0.75 (0.29)	H (2) = 28.033	0.000 **	ε^2^ = 0.010
Benzodiazepines	% Patients tested	21.7	14.3	13.5	χ^2^ (2) = 1.013	0.603	V = 0.106
	Average per patient	2.10 (2.18)	0	2.20 (1.30)	H (2) = 2.827	0.243	ε^2^ = 0.001
	Positive ratio	0.69 (0.46)	0	0.40 (0.55)	H (2) = 1.296	0.255	ε^2^ = 0.000
% Patients that dropped out of treatment		40.1	34.4	13.3	215.46	0.000	V = 0.280

Abbreviations: *d.f.*—degrees of freedom; SD—Standard Deviation; H—Kruskal–Wallis; V—Cramer’s V; ** *p*-value ≤ 0.05.

## Data Availability

Database should be request to the correspondence author.

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
