# Peer review of "Changes in the Care Activity in Addiction Centers with Dual Pathology Patients during the COVID-19 Pandemic"

_jcm, 2022, doi:10.3390/jcm11154341_

Round 1
Reviewer 1 Report
The work “Changes in the care activity in addiction centers with dual pathology patients during the COVID-19 pandemic” is an interesting paper and allows for a comparison of the evolution of addictology demands before and during the pandemic
This study started in 2019 and opportunely it allows to compare data of requests for care before, during and after the lockdown.
Nevertheless will it other channels of requests, such as by phone or through other channels, such as associations, be taken into account, as face-to-face visits to centers were very limited? How were these requests accounted for?
Is it possible to know a little more about the classification system used?
Describe the ethical approach taken (ethics committee or personal protection committee) for the research; how ware participants informed about the study, and the feedback provided.
Can the data be compared to other countries and/or regions?
Also, is it possible to describe how people were assessed on clinical dimensions? For example, how was personality assessed? What assessment instruments were used?
It is necessary to emphasize the interest and clinical perspectives of this work.
Author Response
Dear reviewers,
First of all, we would like to thank you for reading this paper, using your time to review the article, and offering your advice and suggestions to enhance the quality and comprehension of our manuscript. Next, we are going to address each question with the purpose of improving the paper. Finally, we have proceeded to incorporate the suggested changes to our text.
Will it other channels of requests, such as by phone or through other channels, such as associations, be taken into account, as face-to-face visits to centers were very limited? How were these requests accounted for?
Due to the pandemic, most of Andalusian centers used the telephone for admissions to treatment, as well as follow-up. Critically ill patients received face-to-face care from therapists. After confinement due to COVID-19, telephone services have been maintained for patient follow-up. The requests are recorded by health professionals in the Electronic Health Record (EHR). This information has been included in the MS:
“Due to the pandemic, most of the Andalusian centers used the telephone as the main channel for treatment admissions and follow-up. Critically ill patients received face-to-face care from professionals, while telephone services have been maintained for patient follow-up after the confinement period. The requests are recorded by health professionals in the Electronic Health Record (EHR).”
Is it possible to know a little more about the classification system used?
Patients were assigned using Richard Ries's (1992) dimensional model, which is based on the severity and course of the mental and addictive disorder. The diagnosis of addictive and other mental disorders was made by the clinicians of the centers according to ICD-10 criteria. The following clarification has been included in the text:
“The final sample consisted of 2,785 patients diagnosed with an addictive disorder and another mental disorder according to ICD-10. In addition, all patients of the sample had therapeutic prescriptions to receive care in mental health services according to the Ries [40] classification. This is a dimensional model based on the severity of the addictive disorder and other mental disorders. Depending on the severity levels of these disorders, patients receive treatment exclusively in mental health (severe mental disorder and mild addictive disorder), in addiction centers (severe addictive disorder and mild mental disorder) or in both services in a coordinated manner (severe mental health and addictive disorder).”
Describe the ethical approach taken (ethics committee or personal protection committee) for the research; how ware participants informed about the study, and the feedback provided.
This research has been approved by the Research Ethics Committee of the Andalusian Ministry of Health, which certified the compliance with the requirements for the ethical handling of the information.
To access the EHRs, the researchers made a request to the General Secretary of Social Services of the Department of Equality and Social Policies of the Regional Government of Andalusia (Spain). Patients gave their authorization so the information can be registered in the system. The database is fully anonymized for both patients and professionals so it was not possible to inform the participants about the study. The storage and encoding of this data comply with the General Health Law of April 25, 1986 (Spain) and Law 41/2002 of November 14 on patient autonomy, rights, and obligations regarding clinical information and documentation. It also complies with the Organic Law 3/2018 of December 5, 2018, on protecting personal data and guaranteeing digital rights, adapted to European regulations.
This information has been included in the MS.
“This research has been approved by the Research Ethics Committee of the Andalusian Ministry of Health, who certified compliance with the requirements for the ethical handling of the information.
To access the EHRs, the researchers made a request to the General Secretary of Social Services of the Department of Equality and Social Policies of the Regional Government of Andalusia (Spain). Patients gave their authorization so the information can be registered in the system. The database is fully anonymized for both, patient and professional, so it’s not possible to inform the participants about the study. The storage and encoding of this data comply with the General Health Law of April 25, 1986 (Spain) and Law 41/2002 of November 14 on patient autonomy, rights, and obligations regarding clinical information and documentation. It also complies with the Organic Law 3/2018 of December 5, 2018, on protecting personal data and guaranteeing digital rights, adapted to European regulations.”
Can the data be compared to other countries and/or regions?
The admission of patients with substance use disorder is based on the Treatment Demand Indicator Standard of the European Monitoring Centre for Drugs and Drug Addiction. The assessment of addictive and other mental disorders was performed following the ICD 10 classification of disorders. In this sense, both criteria are common to other published studies on dual pathology.
It is specific to this study in that it only includes patients who require coordinated care between specific mental health centers and addiction centers. In this sense, it is possible that this study includes patients with more severe diagnoses of addictions and other mental disorders.
This information has been included as a limitation.
“On the other hand, it is necessary to keep in mind that the study included patients with a high severity of the addictive disorder and other mental disorders, and not only patients with other comorbid disorders. As a consequence, it is likely that the prevalence of dual pathology observed in this study is lower than that observed in other studies of dual pathology conducted in addiction centers.”
Also, is it possible to describe how people were assessed on clinical dimensions? For example, how was personality assessed? What assessment instruments were used?
We appreciate this comment.
We believe that we have not adequately clarified in the MS what the Ries classification (1992) and the diagnostic procedure is used for. All patients were diagnosed by clinicians following the diagnostic criteria of the International Statistical Classification of Diseases and Related Health Problems (ICD-10). To specifically assess personality disorders, the IPDE is available to clinicians.
Assignment to the therapeutic center (addiction centers exclusively, mental health centers exclusively, or coordinated care between the two) was made according to the Ries classification (1992).
We have modified the text and hope that the information is now more understandable.
“The final sample consisted of 2,785 patients diagnosed with an addictive disorder and another mental disorder according to ICD-10. In addition, all patients of the sample had therapeutic prescriptions to receive care in mental health services according to the Ries [40] classification.”
It is necessary to emphasize the interest and clinical perspectives of this work.
We thank the reviewer for this suggestion. We have included the following text:
“Despite these limitations, the present study provides useful information for understanding the changes produced by the COVID-19 pandemic. In particular, our results provide relevant knowledge about a large sample of patients with dual diagnosis and the health care provided in several addiction centers. As this is a coordinated treatment modality, we have observed only the care that has occurred in addiction centers and not the care that these patients have received in mental health centers. Bearing this in mind, the data have shown a reduction in the healthcare received by these patients. Moreover, it is striking that after confinement, the number of patients with dual pathology has decreased. Therefore, it is likely that there is a group of patients with dual pathology who are presently either only receiving care in mental health centers or are not attending health services. Thus, we suggest that the coordinated treatment modality followed by these patients with dual pathology has proven to be insufficient for adequate clinical care in these pandemic times. Therefore, we believe that it is now more necessary than ever to integrate mental health and addiction services for the coordinated treatment of these patients with dual pathology.
Future studies should continue to provide information on care activity and confirm the results found with these patients, so that these data can be used to inform the development of effective and efficient treatments for patients with dual pathology. In addition, future analyses could identify factors that may mediate and prevent some of the major risks in similar situations.”
Reviewer 2 Report
Thank you for the opportunity to review the paper entitled: Changes in the care activity in addiction centers with dual pathology patients during the COVID-19 pandemic. Which presents the important topic of changes in addiction assistance during the COVID-19 pandemic.
Nevertheless, I suggest the authors make a few changes:
- please ensure that in-text citations made in accordance with the Instructions for Authors;
- hypotheses or research questions are missing after the purpose of the paper;
- line 89: please title the Material and Methods;
- Describe the inclusion criteria in detail;
- Figure 1 is unreadable - please include a better resolution (vertical perhaps?);
- please put explanations of abbreviations under the tables;
- values to the square please give in superscript, e.g. line 164;
- the name of the disease entity COVID-19 is written as an acronym, please correct, e.g. line 223;
- the discussion should better correspond with the results of own research - please expand these sections;
- please clearly separate the sections on strengths and weaknesses of the study and conclusions from the discussion - please separate them and title them separately.
Kind regards.
Author Response
Dear reviewers,
First of all, we would like to thank you for reading this paper, using your time to review the article, and offering your advice and suggestions to enhance the quality and comprehension of our manuscript. Next, we are going to address each question with the purpose of improving the paper. Finally, we have proceeded to incorporate the suggested changes to our text.
1) Please ensure that in-text citations made in accordance with the Instructions for Authors
Following your suggestions, we have revised the MS according to the following “Instructions for authors”.
2) Hypotheses or research questions are missing after the purpose of the paper
We welcome your suggestions. We have proceeded to include the following hypotheses:
“As hypotheses based on those objectives, it is expected that:
- the evolution of admissions to treatment decreased during confinement.
- patients with dual pathology who attend addiction care centers present changes in their sociodemographic, consumption, and diagnosis profile during the pandemic compared to the previous period.
- care indicators related to therapeutic appointments, toxicological tests, and treatment abandonment will change during the pandemic compared to the previous period.”
3) line 89: please title the Material and Methods;
We now include this section in the MS.
4) Describe the inclusion criteria in detail;
Following your suggestions, we have now specified the inclusion criteria in more detail.
“Inclusion criteria were the following: 1) to be older than 18 years; 2) to have at least one diagnosis according to the International Classification of Diseases 10 (ICD-10) of an addictive disorder (cocaine, heroin, alcohol, cannabis or pathological gambling) and another comorbid mental disorder; and 3) to have a clinical indication to receive coordinated care with mental health services.”
5) Figure 1 is unreadable - please include a better resolution (vertical perhaps?);
Following your suggestions, we have proceeded to modify the resolution of the figure
6) Please put explanations of abbreviations under the tables;
Following your suggestions, we have proceeded to explain the abbreviations
7) values to the square please give in superscript, e.g. line 164;
Done. Thanks.
8) the name of the disease entity COVID-19 is written as an acronym, please correct, e.g. line 223;
We have correct this. Thanks
9) the discussion should better correspond with the results of own research - please expand these sections; - please clearly separate the sections on strengths and weaknesses of the study and conclusions from the discussion - please separate them and title them separately.
We appreciate these suggestions and have now incorporated changes to the discussion accordingly.
Round 2
Reviewer 2 Report
I recommended publication in current form.
Thank you.